# iPSC-Derived Endothelial Cells Reveal LDLR Dysfunction and Dysregulated Gene Expression Profiles in Familial Hypercholesterolemia

**DOI:** 10.3390/ijms25020689

**Published:** 2024-01-05

**Authors:** Irina S. Zakharova, Alexander I. Shevchenko, Mhd Amin Arssan, Aleksei A. Sleptcov, Maria S. Nazarenko, Aleksei A. Zarubin, Nina V. Zheltysheva, Vlada A. Shevchenko, Narek A. Tmoyan, Shoraan B. Saaya, Marat V. Ezhov, Valery V. Kukharchuk, Yelena V. Parfyonova, Suren M. Zakian

**Affiliations:** 1Federal Research Centre Institute of Cytology and Genetics, Siberian Branch of the Russian Academy of Sciences, 630090 Novosibirsk, Russia; zakharova@bionet.nsc.ru (I.S.Z.); epigene@bionet.nsc.ru (A.I.S.); amin.arssan19@gmail.com (M.A.A.); n.zheltysheva@alumni.nsu.ru (N.V.Z.); vlada.a.shevchenko@gmail.com (V.A.S.); 2Research Institute of Medical Genetics, Tomsk National Research Medical Centre, Russian Academy of Science, 634050 Tomsk, Russia; alexei.sleptcov@medgenetics.ru (A.A.S.); maria-nazarenko@medgenetics.ru (M.S.N.); aleksei.zarubin@medgenetics.ru (A.A.Z.); 3Federal State Budgetary Institution, National Medical Research Centre of Cardiology Named after Academician E.I. Chazov, Ministry of Health of Russian Federation, 121552 Moscow, Russia; ntmoyan@gmail.com (N.A.T.); marat_ezhov@mail.ru (M.V.E.); v_kukharch@mail.ru (V.V.K.); yeparfyon@mail.ru (Y.V.P.); 4E.N. Meshalkin National Medical Research Centre, Ministry of Health Care of the Russian Federation, 630055 Novosibirsk, Russia; shoraans@gmail.com

**Keywords:** familial hypercholesterolemia, LDLR, patient-specific iPSCs, directed differentiation, endothelial cells

## Abstract

Defects in the low-density lipoprotein receptor (LDLR) are associated with familial hypercholesterolemia (FH), manifested by atherosclerosis and cardiovascular disease. LDLR deficiency in hepatocytes leads to elevated blood cholesterol levels, which damage vascular cells, especially endothelial cells, through oxidative stress and inflammation. However, the distinctions between endothelial cells from individuals with normal and defective LDLR are not yet fully understood. In this study, we obtained and examined endothelial derivatives of induced pluripotent stem cells (iPSCs) generated previously from conditionally healthy donors and compound heterozygous FH patients carrying pathogenic *LDLR* alleles. In normal iPSC-derived endothelial cells (iPSC-ECs), we detected the LDLR protein predominantly in its mature form, whereas iPSC-ECs from FH patients have reduced levels of mature LDLR and show abolished low-density lipoprotein uptake. RNA-seq of mutant *LDLR* iPSC-ECs revealed a unique transcriptome profile with downregulated genes related to monocarboxylic acid transport, exocytosis, and cell adhesion, whereas upregulated signaling pathways were involved in cell secretion and leukocyte activation. Overall, these findings suggest that LDLR defects increase the susceptibility of endothelial cells to inflammation and oxidative stress. In combination with elevated extrinsic cholesterol levels, this may result in accelerated endothelial dysfunction, contributing to early progression of atherosclerosis and other cardiovascular pathologies associated with FH.

## 1. Introduction

The low-density lipoprotein (LDL) receptor plays an important role in human lipid metabolism [1]. Numerous mutations within the *LDLR* gene are pathogenic and cause hereditary familial hypercholesterolemia (FH). Pathogenic *LDLR* alleles have been found in approximately 85–90% of patients with FH [2,3]. According to the latest data, FH has a prevalence of 1/200 in the heterozygous form and is much less frequent at 1/300,000 in the homozygous form (including compound heterozygotes) [4,5]. Patients with FH have elevated blood cholesterol levels and concomitant lipid metabolism diseases, such as atherosclerosis, coronary heart disease, and Alzheimer’s disease [6].

The main role in FH pathogenesis is attributed to LDL receptors in liver cells, responsible for cholesterol uptake and its subsequent metabolism into bile acids [7,8,9]. However, when the LDL receptors are defective or absent, hepatocytes have impaired LDL internalization, ultimately leading to increased cholesterol levels in the bloodstream [3,10]. Excess cholesterol can be deposited as xanthomas on the skin, corneal arcus, feet, and hand tendons [11]. Elevated blood cholesterol also affects blood vessels and vascular cells through the lipid accumulation inside the intima, causing oxidative stress, inflammatory cell recruitment, and local cytokine production [7,8,9,12,13,14]. This leads to endothelial dysfunction and atheroma formation in the arteries, aortic root, and valves, followed by more severe pathologies for example myocardial infarction [3,11,15,16,17,18].

Although the endothelium is implicated in FH pathogenesis, little is known about how vascular endothelial cells per se differ between individuals with normal and pathogenic *LDLR* alleles. Our study focused on identifying LDLR protein expression and transcriptomic patterns in endothelial cells with pathogenic *LDLR* gene mutations to elucidate how this may be related to the FH progression.

The current trend in human disease modelling is to generate patient-specific iPSCs from individuals with hereditary pathologies and to elucidate the mechanisms of this pathology at the molecular and cellular level using relevant cells obtained by iPSC differentiation [19]. Patient-specific iPSC-derived endothelial cells (iPSC-ECs) are widely used to model various disorders, including pulmonary arterial hypertension, Moyamoya disease, fibrodysplasia ossificans progressiva, Huntington’s disease, Kawasaki’s disease, type I diabetes mellitus, atrial or ventricular septal defects, pulmonary valve stenosis, and ventricular septal defects, cardiomyopathy, calcified aortic valve disease, hemophilia A, diabetic endotheliopathy, Hutchinson–Gilford progeria syndrome, autosomal dominant cerebral arteriopathy with subcortical infarcts and leukoencephalopathy, and peripheral artery disease [20,21,22,23,24,25].

There are known lines of iPSCs from patients with FH and examples of their use to model this pathology, which have only been carried out on hepatocyte-like derivatives of iPSCs [26,27,28,29,30]. Despite the knowledge that endothelial cells are the main target in atherosclerosis and the progression of cardiovascular pathology in FH, ECs, or iPSC-derived ECs from patients with FH have not yet been obtained or studied. Current studies of endothelial dysfunction in FH patients are based on blood biomarkers, blood flow, and microvascular function indicators [31,32,33].

In endothelial cells, LDLR is thought to be involved in the endocytosis and transcytosis of circulating LDLs [34,35,36]. LDLR is expressed in the blood–brain barrier endothelium [37] and in human umbilical artery endothelial cells (HUAECs) [34]. However, we found no information on LDLR expression in endothelial cells from FH patients, including iPSC-ECs. With this work, we fill this gap and hope that studying patient-specific iPSC-ECs may potentially contribute to the understanding of FH pathogenesis and aid in the development of new treatment approaches.

We reasoned that iPSC-derived endothelial cells lacking both native *LDLR* alleles would have more pronounced and potent effects of LDLR dysfunction and would therefore be more informative for studying gene function. Thus, to model the cellular and molecular mechanisms of FH in iPSC-ECs, we used two previously obtained lines of induced pluripotent stem cells (iPSCs) from patients who were compound heterozygotes for pathogenic and likely pathogenic allelic variants of the *LDLR* gene [38,39,40]. We obtained endothelial derivatives by direct differentiation of iPSC lines from a conditionally healthy donor and from patients with FH. When we compared the resulting iPSC-ECs for LDLR protein expression and their transcriptomes, we found that endothelial cells with damaged LDLR are themselves more prone to dysfunction, oxidative stress, and chronic inflammation, which may underlie the earlier and accelerated progression of atherosclerosis and cardiovascular disease in patients with FH.

## 2. Results

### 2.1. Generation and Characterization of Endothelial Derivatives from iPSCs

To study the properties of endothelial cells with pathogenic LDLR allelic variants, we obtained endothelial derivatives via directed differentiation from iPSC lines FH 1.3.1S and FH 3.2.8T of patients with FH as well as from iPSC lines K6-4f and K7-4Lf of conditionally healthy donors. On day 8, we assessed the directed differentiation efficiency by quantifying VE-cadherin-positive cells by FACS. We found no significant differences between cultures of iPSC-ECs in the number of VE-cadherin-positive cells, which accounted for more than 90% of the differentiated derivatives (Figure 1a). This indicates that the endothelial differentiation of iPSCs from patients with FH was as efficient as the differentiation of iPSCs from a healthy donor. All resulting derivatives show an immunophenotype with the surface antigen CD31 and von Willebrand factor, attributable to endothelial cells (Figure 1b and Appendix A). Furthermore, iPSC-ECs with and without the *LDLR* mutation have similar ability to form capillary-like structures in Matrigel without significant differences in angiogenic potential parameters (Figure 1c–f). However, when we assessed LDL uptake, we found that iPSC-ECs from patients with FH were significantly reduced in their ability to internalize the fluorescently labelled Dil-LDL (Figure 2).

### 2.2. iPSCs and iPSC-Derived Endothelial Cells (iPSC-ECs) from Patients with FH Have Reduced Levels of Mature LDLR

LDL receptors have been found not only in hepatocytes but also in the other cell types, including iPSCs and endothelium [26,27,28,34,37]. Our results, demonstrating the ability of iPSC-ECs to uptake LDL, indicate that these cells have LDL receptors. We further investigated the presence of mature and immature LDLR forms in protein extracts from iPSCs and iPSC-ECs with normal and pathogenic LDLR alleles using immunoblotting. As a control, we used HepG2 hepatocyte-like cells with characteristics of mature hepatocytes derived from a patient with hepatocarcinoma [41]. We found that mature LDLR was predominantly detected in iPSCs and iPSC-ECs with the native *LDLR* gene, as well as in hepatocyte-like HepG2 cells (Figure 3a–c). The relative LDLR protein levels in hepatocyte-like HepG2 cells, both normal iPSCs and iPSC-ECs, did not differ significantly from each other. However, a significantly reduced level of the mature LDLR was found in iPSCs and iPSC-ECs from FH patients compared to normal iPSCs, their endothelial derivatives, and the hepatocyte-like HepG2 cell line (Figure 3a–c). In the iPSC line FH 1.3.1 and its endothelial derivatives, the level of immature LDLR was significantly increased compared to the mature LDLR. In the iPSC line FH 3.2.8 and its endothelial derivatives, both LDLR forms are detected at low levels (Figure 3a–c), and the total LDLR level is significantly reduced (Figure 3c). In the iPSC line FH 1.3.1 and its endothelial derivatives, the total LDLR level does not differ from that of conditionally healthy iPSCs and their endothelial derivatives (Figure 3b). Thus, we have shown that both iPSCs obtained from FH patients and their endothelial derivatives have reduced levels of mature LDLR.

### 2.3. Transcriptome Profiling Reveals Dysregulation of Several Signaling Pathways in LDLR Mutant iPSC-ECs

We performed bulk RNA sequencing (RNA-seq) on two control iPSC-EC cultures (CTRL) and two LDLR mutant iPSC-EC cultures (FH), using three technical replicates for each experiment. The PCA plot of the transcriptome of 12 samples showed a clustering of three technical replicates together and a distinct distribution in iPSC-EC cultures according to disease by dimension 2 (Figure 4a). Direct comparison of total RNA expression of iPSC-ECs between CTRL and FH groups revealed 39 differentially expressed genes (DEGs), of which 26 were downregulated and 13 were upregulated in FH (logFC |1|, FDR < 0.05; Figure 4b,c, and Appendix A). The top five down-regulated genes were *MEG3*, *MEG8*, *FGB*, *MIR381HG*, and *HSALNG0039225*, whereas the top five over-expressed genes were *LINC01291*, *LINC02968*, *HSALNG0024559*, *IGLON5*, and *ILDR2* (Figure 4b). The average expression level (transcripts per million) of the *LDLR* gene was 130 and 115 in the FH and CTRL iPSC-EC lines, respectively. The average expression level (transcripts per million) of the *PCSK9* gene was 0.6 and 0.4 in FH and CTRL iPSC-EC lines, respectively.

Gene ontology analysis of DEGs revealed several dysregulated biological pathways in *LDLR* mutant iPSC-EC lines (Figure 4d, Appendix A). Monocarboxylic acid transport (GO:0015718) genes (*PLA2G5*, *SLC16A6*, *ADORA2B*, and *ABCC4*) were downregulated in *LDLR* mutant iPSC-EC lines. Downregulated genes are also involved in the biological processes (BPs) of regulating exocytosis (GO:0017157; *ADORA2B*, *FGB, STXBP6*, *PLA2G5*, *ABCC4*, and *SEMA5A*), cell–cell adhesion (GO:0098609; *FGB*, *SPARCL1*, *STXBP6*, and *CNTN4*), and positive regulation of the response to external stimuli (GO:0032103; *ADORA2B*, *PLA2G5*, *SEMA5A*, *S100A14*, and *HLA-DQB1*).

The 13 upregulated DEGs were involved in the BPs that control cell secretion (GO:0046903; *XDH*, *SYT11*, and *ILDR2*) and regulate leukocyte activation (GO:0002694; *SYT11*, *SIRPA*, and *ILDR2*).

Among these DEGs, three were related to angiogenesis (GO:0001936) genes such as *XDH*, *SEMA5A*, and *SULF1*. The expression of *SEMA5A* and *SULF1* was downregulated in FH iPSC-ECs, whereas *XDH* was upregulated.

Reactome Gene Sets analysis showed that downregulated DEGs such as *HLA-DQB1*, *IFIT2*, and *UBA7* were related to the interferon signaling pathway (R-HSA-913531). In terms of KEGG analysis, the DEGs were mainly involved in the Rap1 signaling pathway (hsa04015), including downregulated *ADORA2B*, *GNAO1*, and *SULF1* but the upregulated *EFNA2* gene.

STRING analysis revealed an interaction between LDLR and SYT11, CNTN4, and SEMA5A at the protein level (Figure 4e).

## 3. Discussion

In our study, we were the first to obtain iPSC-ECs from compound heterozygous patients with FH and demonstrated dysfunction of the LDLR protein due to its impaired maturation. However, the pattern of LDLR abnormalities appears to vary depending on the mutations and their combination, as different functional domains of the protein are affected. The LDLR mutations in the FH 1.3.1 patient are c.530C>T, leading to a p.Ser177Leu substitution (ClinVar ID 3686), and c.1054T>C, resulting in a p.Cys352Arg substitution (ClinVar ID 251618). The first is considered pathogenic, the second as likely pathogenic, and both are transport-defective mutations that cause the immature LDLR to be retained in the endoplasmic reticulum and not transported to the Golgi apparatus for further maturation [42,43,44]. In patient FH 3.2.8, an extended deletion c.2141-966_2390-330del spans introns 14–16 and exons 15–16 and is considered to be pathogenic according to bioinformatic predictions, as it interferes with the production of the mature LDLR protein [40,45,46]. The second mutation, c.1327T>C to p.Trp443Arg (ClinVar ID 998052), which is considered likely pathogenic, causes a recycling defect, in which ligands are not released from the complex with LDLR in endosomes, and, as a result, LDLR is not recycled to the cell surface [38,40,45,46]. Our results on LDLR expression allow us to conclude that all LDLR mutations in theses iPSC lines and their iPSC-EC derivatives lead to a reduction in the mature form of the LDLR protein, whereas the combination of mutations in the second patient significantly reduces both mature and immature LDLR forms.

Some LDLR protein defects can be corrected by drug therapy in experiments. Recently, a pharmacological blockade of endoplasmic reticulum-associated degradation or treatment with pharmacological chaperones has been found to allow transport of immature LDLR to the cell membrane surface, restoring receptor function despite its incomplete maturation [42]. Thus, FH-modelled iPSCs and iPSC-ECs could be a reliable tool to further aid the development and screening of targeted drugs to treat the consequences of specific mutations in patients with FH.

In this study, we also compared iPSC-ECs with normal and defective LDLR at the transcriptomic level. We found no difference in *LDLR* gene expression between FH and CTRL iPSC-ECs, suggesting that LDLR function is mainly abolished at the post-transcriptional level, as also shown by immunoblotting results.

Transcriptome profiling revealed the downregulation of the most genes (67%) in *LDLR* mutant iPSC-ECs. We found that biological processes related to monocarboxylic acid transport, exocytosis, and cell–cell adhesion were downregulated in LDLR mutant iPSC-EC lines. In contrast, cell secretion and leukocyte activation were upregulated. Thus, the quantitative reduction in mature LDLR in iPSC-ECs abolished endothelial function, which was manifested both in the decreased ability to uptake LDL and at the transcriptomic level. Of special note, the best known FH-related hit, the *PCSK9* gene, which promotes LDLR degradation and plays a critical role in regulating cholesterol homeostasis [47,48], shows no significant differences in expression between control and FH iPSC-ECs, and patients carry no pathogenic mutations in the gene (clinical genotyping data).

The downregulated expression of monocarboxylic acid transporters reflects a dysregulation of cellular metabolism in LDLR-deficient iPSC-ECs. These transporters were directly and indirectly linked with metabolites acetyl-CoA and fatty acids (organic anions), quantitative variations of which can affect chromatin and gene expression regulation [49,50,51], explaining the observed differences in transcriptome between control and mutant *LDLR* iPSC-ECs. It should also be noted that metabolic pathways in cells and organisms are interconnected, so it cannot be excluded that changes in the monocarboxylic acid transporters may indirectly affect other pathways involved in transcriptional regulation. In addition, decreased levels of antioxidant transport associated with the transporters could potentially render cells more susceptible to oxidative stress [52].

We also found dysregulation of angiogenesis genes (*XDH*, *SEMA5A*, and *SULF1*) and the Rap1 pathway (*ADORA2B*, *GNAO1*, *SULF1*, and *EFNA2*), which promotes endothelial homeostasis and may be involved in endothelial dysfunction-associated cardiovascular pathologies [53]. However, the iPSC-ECs with normal and defective LDLR showed no differences in the ability to form capillary-like structures in an angiogenesis test, both in terms of total vessel length and number of branching points. Therefore, it is likely that the identified dysregulation in angiogenesis genes may be related to other aspects of this process. It is also noteworthy that Rap1 signaling and the *XDH* gene are both related to oxidative stress [54,55]. In addition, the atheroprotective role of Rap1 in endothelial cells was recently shown in a FH mouse model, which consists of limiting the transmission of proinflammatory cytokines and preventing the expression of inflammatory receptors [56]. Thus, the reduced level of Rap1 expression that we found in iPSC-ECs from FH patients may enhance pro-inflammatory signaling and contribute to the progression of atherosclerosis.

Additionally, the reconstructed protein–protein interaction network showed an association between LDLR and some molecules such as SYT11, CNTN4, and SEMA5A. Interestingly, semaphorin and ephrin family members are involved in axon guidance and synaptic plasticity, and they are also important in endothelial cell–leukocyte interactions during atherogenesis [57]. Synaptotagmin-11 (SYT11) and contactin 4 (CNTN4) are both essential for neural development [58,59]. SYT11 is a vesicle trafficking protein that can suppress microglial activation by inhibiting cytokine secretion and phagocytosis [60]. CNTN4, a crucial adhesion protein, is involved in T-cell activation and oxLDL-induced cell apoptosis and inflammation in THP-1 macrophages [61,62]. Thus, our data suggest that dysregulation of neuronal guidance molecules (chemoattraction and chemorepulsion) can be considered as a possible mechanism for endothelial dysfunction in LDLR mutant iPSC-EC derivatives.

Overall, the transcriptomic analysis of iPSC-ECs with defective LDLR identified a number of genes and signaling pathways that could potentially be important for the progression of endothelial dysfunction in FH patients. Violations in endothelial cells that we found in FH patient-specific iPSC-ECs, such as impaired intracellular transport, cell–cell contacts, and oxidative stress, are also known aspects of endothelial dysfunction in atherosclerosis [9,13,63,64,65].

The mechanisms that trigger endothelial dysfunction in FH are not fully understood [66]. Typically, endothelial dysfunction in FH is considered to be a consequence of abnormalities in the response to the elevation and retention of cholesterol in plasma [67,68]. Nevertheless, recent evidence suggests that other processes unrelated to cholesterol levels, its retention and oxidation may also lead to endothelial dysfunction in FH. For example, red blood cells with defective LDLRs exacerbate endothelial dysfunction in FH patients [66]. In line with this, our data suggest that endothelial cells with damaged LDLRs per se appear to be predisposed to dysfunction, oxidative stress, and chronic inflammation, which, in combination with elevated extrinsic LDL cholesterol levels, may accelerate endothelial dysfunction, contribute to atherogenesis, and cause early and accelerated FH progression. Indirect evidence that ECs may be predisposed to dysfunction in FH patients is provided by the study showing that abnormalities in endothelial functioning, verified by flow-mediated dilation, were detected in children with FH long before the appearance of structural atherosclerotic vascular changes [69]. However, further detailed studies are needed to confirm and understand our findings.

## 4. Materials and Methods

### 4.1. Ethical Statements

The patient-specific iPSC lines used in this work, with the hPSCreg numbers ICGi022-A, ICGi036-A, and ICGi038-A, were previously derived in compliance with all ethical standards [38,39,70].

### 4.2. iPSC Lines and Their Cultivation

Two patient-specific iPSCs cell lines, called FH 1.3.1S and FH 3.2.8T, hPSCregs numbers ICGi036-A and ICGi038-A, were used in the study [38,39]. They were obtained previously from FH patients who were compound heterozygotes with pathogenic and likely pathogenic variants of both LDLR alleles, namely, p.Ser177Leu/p.Cys352Arg (dbSNP IDs: rs121908026/rs879254769) for FH 1.3.1S and p.Glu714_Ile796del/p.Trp443Arg (dbSNP IDs: no rs number/rs773566855) for FH 3.2.8T [38,39,40]. The control was the iPSC lines K7-4Lf and K6-4f from healthy donors, hPSCreg number ICGi022-A and ICGi021-A, respectively [70]. The iPSCs were cultured in E8 medium (Thermo Fisher Scientific, Waltham, MA, USA) on a surface coated with Matrigel matrix (Corning, New York, NY, USA) in an incubator at 37 °C and 5% CO_2_. During re-plating, cells were disaggregated with 0.5 mM EDTA (Thermo Fisher Scientific, Waltham, MA, USA) and seeded at a 1:4 ratio in E8 medium (Thermo Fisher Scientific, Waltham, MA, USA) supplemented with 10 mM Rho-kinase inhibitor thiazovivin (Merck, Darmstadt, Germany).

### 4.3. Directed Differentiation of iPSCs into Endothelial Derivatives

Directed differentiation of iPSCs obtained from a healthy donor and patients with FH in the endothelial direction was carried out according to the protocol [71] with our modifications [72]. The protocol includes three steps: mesodermal differentiation, endothelial differentiation, and selection of endothelial derivatives by magnetic sorting. In the first step (mesodermal differentiation), iPSCs were seeded in the E8 medium (Thermo Fisher Scientific, Waltham, MA, USA) on a surface treated with Matrigel (Corning, New York, USA) to achieve 60–70% cell confluence the next day. On day 2, we washed the iPSCs from the E8 medium (Thermo Fisher Scientific, Waltham, MA, USA) with PBS (Rosmedbio, Saint Petersburg, Russia) and added new medium containing RPMI 1640 (Thermo Fisher Scientific, Waltham, MA, USA), 6 μM Chir99021 (Selleckchem, Houston, TX, USA), 1% B27 supplement without insulin (Thermo Fisher Scientific, Waltham, MA, USA), and 100 units/mL penicillin-streptomycin (Thermo Fisher Scientific, Waltham, MA, USA) to initiate mesodermal differentiation. On day 3, the culture medium was completely replaced with a similar one, containing RPMI 1640, 1% B27 without insulin, and 100 units/mL penicillin-streptomycin (all from Thermo Fisher Scientific, Waltham, MA, USA), with the concentration of Chir99021 (Selleckchem, Houston, TX, USA) halved to 3 μM. The endothelial differentiation step was performed from day 5 to day 8. The growth medium from the previous step was completely removed, the cells were washed with PBS (Rosmedbio, Saint Petersburg, Russia) and EGM-2 complete endothelial medium (Lonza, Basel, Switzerland) supplemented with 50 ng/mL VEGF and 25 ng/mL bFGF (all from Sci-store, Moscow, Russia), and 10 μM SB431542 (R&D Systems, NE Minneapolis, MN, USA) was added. At the final stage of endothelial differentiation (day 9), the culture was enriched by magnetic sorting, selecting cells based on the CD31 marker of mature endothelial cells. Magnetic beads conjugated to anti-CD31 antibodies (Miltenyi Biotec, Bergisch Gladbach, Germany) were used for sorting according to the manufacturer’s protocol. Prior to incubation with magnetic beads, cells were disaggregated using Accutase enzyme (Thermo Fisher Scientific, Waltham, MA, USA) and passed through a 0.70 μm cell strainer to remove conglomerates. Sorted CD31-positive cells were seeded in complete endothelial medium EGM-2 (Lonza, Basel, Switzerland) on a culture surface coated with type 4 collagen (Merck, Darmstadt, Germany). Endothelial derivatives were passaged every 6–7 days using TrypLE Express (Thermo Fisher Scientific, Waltham, MA, USA) at a ratio of 1:2–1:3, depending on cell density.

### 4.4. Flow Cytometry

Quantitative detection of the endothelial marker VE-cadherin (CD144) in cultures after differentiation was carried out by flow cytometry using a FACS Aria III instrument at the Centre for Analysis of Biological Objects in the Institute of Cytology and Genetics, SB RAS. We used an anti-VE-cadherin (CD144) antibody (BD, Franklin Lakes, NJ, USA) and an isotype control mouse IgG1 (BD, Franklin Lakes, NJ, USA), both conjugated to the PerCP-Cy5.5 fluorochrome (Table 1). Cells were disaggregated using Accutase (Thermo Fisher Scientific, Waltham, MA, USA) to avoid damage to the cell membrane containing the surface marker. Staining of endothelial derivatives with antibodies and appropriate isotype controls was performed according to the manufacturer’s protocol (BD, Franklin Lakes, NJ, USA). For one reaction, 10^5^ cells were used in 100 μL of 1% bovine serum albumin (BSA) (Merck, Darmstadt, Germany) in PBS (Rosmedbio, Saint Petersburg, Russia) with the addition of 5 μL of antibodies or 2 μL of isotype control. The experiment was performed in three biological and three technical replicates. In each experiment, 10^4^ events were counted.

### 4.5. Immunostaining

The immunostaining procedure was described in detail previously [73,74,75]. Briefly, differentiated endothelial derivatives were fixed with 4% PFA (Merck, Darmstadt, Germany) for 10 min, permeabilized with 0.5% Triton X-100 (Merck, Darmstadt, Germany) for 30 min and then blocked with 1% BSA (Merck, Darmstadt, Germany) in PBS (Rosmedbio, Saint Petersburg, Russia). Incubation with primary antibodies was carried out overnight at +4 °C. Secondary antibodies were incubated with cells for 1.5 h in the dark at room temperature. Cell nuclei were stained with DAPI. Imaging was performed on a Ti-E inverted fluorescence microscope equipped with a Nikon DS-Qi1Mc digital sight microscope camera using NIS Elements Advanced Research software version 4.30 (all from Nikon, Tokyo, Japan). A list of primary and secondary antibodies is given in Table 1.

### 4.6. Functional Evaluation of Endothelial Cells

To evaluate angiogenic potential, 200 μL of Matrigel (Corning, New York, NY, USA) diluted 1:1 with EGM-2 medium (Lonza, Basel, Switzerland) was polymerized in a 1 cm^2^ well for 15 min at 37 °C. A suspension of 2 × 10^5^ endothelial cells with EGM-2 medium (Lonza, Basel, Switzerland) was then evenly distributed over the surface of the well and placed in an incubator at 37 °C. Capillary-like structures formed by endothelial cells were detected after 4 h on a Ti-E inverted fluorescence microscope equipped with a Nikon DS-Qi1Mc digital sight microscope camera using NIS Elements Advanced Research software version 4.30 (all from Nikon, Tokyo, Japan). The parameters of capillary-like structures formed by endothelial cells were assessed using the AngioTool program [76].

LDL uptake capacity was assessed by adding fluorescently labelled Dil-LDL (Thermo Fisher Scientific, Waltham, MA, USA) to proliferating endothelial cells. Cells were incubated in serum-free medium for 24 h before staining. Staining was performed overnight in serum-free medium supplemented with Dil-LDL (Thermo Fisher Scientific, Waltham, MA, USA) at a concentration of 5 μg/mL. Imaging was performed using a Ti-E inverted fluorescence microscope (Nikon, Tokyo, Japan) and NIS Advanced Research software version 4.30. To quantify fluorescence intensity, cells were imaged in 10 fields of view at the same exposure value of 400 ms (Texas Red channel), and the images were processed in the ImageJ program (https://ij.imjoy.io/, accessed on 20 October 2023).

### 4.7. Immunoblotting

Proteins were extracted from 10^5^ cells in RIPA buffer (Merck, Darmstadt, Germany). Extract of HepG2 cells were received from the Collective Centre of ICG SB RAS “Collection of Pluripotent Human and Mammalian Cell Cultures for Biological and Biomedical Research” (https://ckp.icgen.ru/cells/; http://www.biores.cytogen.ru/brc_cells/collections/ICG_SB_RAS_CELL, accessed on 1 November 2023). Each sample of 8 μg was separated by 10% SDS-PAGE using a Tetra Mini-Protein Electrophoresis BIORAD system (Bio-Rad Lab, Berkeley, CA, USA). Recombinant human LDLR (R&D Systems, NE Minneapolis, MN, USA) was used as a control. The separated proteins were transferred from the gel to a PVDF membrane (Bio-Rad Lab, Berkeley, CA, USA) using a Mini Trans-Blot wet transfer system (Bio-Rad Lab). The membrane was split into two parts according to a pre-stained protein molecular weight marker (Bio-Rad Lab, Berkeley, CA, USA). Each part of the membrane was precipitated with antibodies, one against the target protein LDLR and the other against the reference protein ACTB. Secondary antibodies against goat and rabbit IgG conjugated to horseradish peroxidase were used for detection. The primary and secondary antibodies and their dilutions are shown in Table 1. Chemiluminescent signal detection was performed using the Bio-Rad Clarity Max Western ECL Substrate Kit on a Bio-Rad ChemiDoc MP instrument. Densitometric analysis to measure differences in protein levels was performed using ImageJ software (https://ij.imjoy.io/, accessed on 20 October 2023). Based on the measurement results, the ratio of signal intensities between the target and reference proteins was obtained. We carried out each immunoblotting experiment in triplicate. Box plots were constructed, and statistical comparisons were performed using the Wilcoxon test with Bonferroni correction for multiple comparisons in the R package (R version 4.2.0; R Core Team (2022). R: A language and environment for statistical computing. R Foundation for Statistical Computing, Vienna, Austria. URL https://www.R-project.org/, accessed on 20 October 2023).

### 4.8. RNA-seq

For RNA sequencing, iPSC-EC cultures were lysed with Trizol reagent, and then total RNA was extracted with a PureLink RNA micro kit (Invitrogen, Carlsbad, CA, USA). RNA quality was assessed using the BioAnalyser and RNA 6000 Nano Kit (Agilent, Santa Clara, CA, USA). Samples with an RNA integrity number > 8.0 were used for cDNA library preparation. PolyA RNA was purified from total RNA samples with Dynabeads^®^ mRNA Purification Kit (Thermo Fisher Scientific, Waltham, MA, USA). The Illumina cDNA library was prepared from polyA RNA using NEBNext^®^ Ultra™ II RNA Library Prep (New England BioLabs, Ipswich, MA, USA) according to the instructions. The resulting cDNA library was sequenced at 75 bp read length on the Illumina HiSeq 1500 platform by Genoanalytica (https://www.genoanalytica.ru, accessed on 01 October 2023). A minimum of 30 million read pairs were generated for each sample.

### 4.9. Statistics

Statistical data processing was performed using the Wilcoxon test with Bonferroni correction for multiple comparisons. Statistical calculations and box plots were made using the R package version 4.2.0 (R version 4.2.0; R Core Team (2022). R: A language and environment for statistical computing. R Foundation for Statistical Computing, Vienna, Austria. URL https://www.R-project.org/, accessed on 20 October 2023).

RNA sequencing data were processed using DRAGEN Bio-IT v.3.9.5 (Illumina) and aligned to a reference human hg38 genome. The quality of the RNA-seq data was assessed using MultiQC v.1.11 [77]. Identification of differentially expressed genes (DEGs) between two control iPSC-EC cultures (CTRL) and two LDLR mutant iPSC-EC cultures (FH) was carried out using the EdgeR tool [78] with the following criteria: (i) log fold change ≥|1| and (ii) FDR adjusted *p*-value < 0.05. Genes with negative fold change values were termed downregulated, whereas those with positive fold change values were termed upregulated. To analyze the gene ontology terms and pathways affected in the LDLR mutant iPSC-ECs, both downregulated and upregulated DEGs were used as input in the online server Metascape [79]. The protein–protein interaction network of LDLR and DEGs was constructed using the STRING database (https://string-db.org, accessed on 1 October 2023).

## 5. Conclusions

In the study, we showed that typical endothelial derivatives of human iPSCs predominantly produced mature LDLR protein, whereas its level is significantly reduced in endothelial derivatives carrying pathogenic *LDLR* alleles. Defective LDLR is manifested at the transcriptomic level as down-regulated expression of monocarboxylic acid transporters. Altered expression was also found for genes involved in exocytosis, cell–cell adhesion, cell secretion, and leukocyte activation. Overall, LDLR-deficient iPSC-ECs revealed a bias in expression in a number of genes and signaling pathways that may be important for the progression of endothelial dysfunction in FH patients. Endothelial cells with damaged LDLR per se appear to be predisposed to dysfunction, oxidative stress, and chronic inflammation, which may facilitate early and accelerated FH progression. However, further detailed studies are needed to confirm and understand our findings.

## Figures and Tables

**Figure 1 ijms-25-00689-f001:**
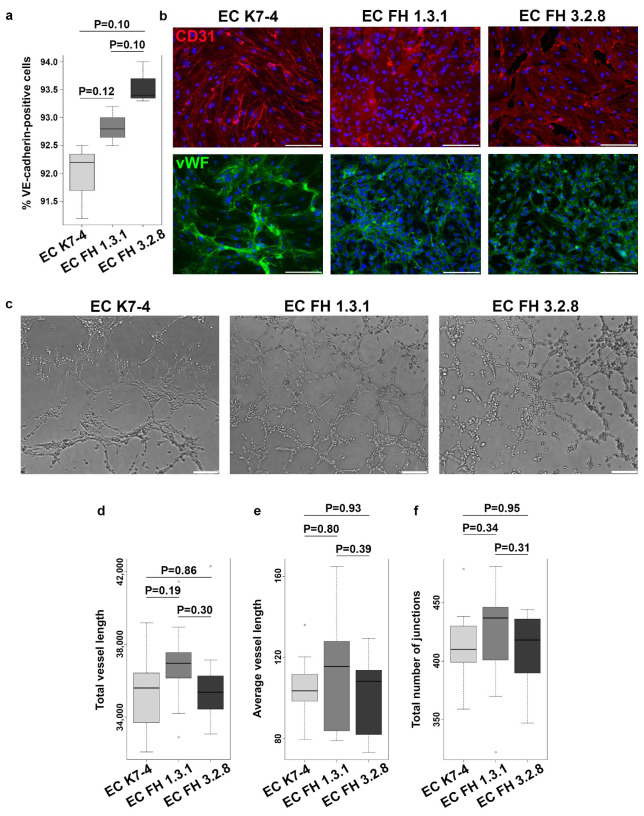
Characteristics of endothelial derivatives obtained by differentiating iPSCs from patients with FH and a healthy donor. (**a**) VE-cadherin quantification of iPSC-derived ECs from a healthy donor (K7-4) and patients with FH (FH 1.3.1S and FH 3.2.8T). The experiment was performed in three biological and three technical replicates. (**b**) iPSC-ECs from a healthy donor and patients with FH are positively stained with antibodies against CD31 (red) and von Willebrand factor (green). Nuclei are stained with DAPI. The scale bar is 100 microns. Magnification 200×. (**c**) Capillary-like structures formed by endothelial derivatives of iPSCs from a healthy donor (EC K7-4) and patients with FH (FH EC 3.1S and FH EC 3.2.8T). The scale bar is 100 microns. Magnification 100×. (**d**–**f**) Quantification of angiogenic potential based on (**d**) total and (**e**) average capillary length, and (**f**) total number of branching points calculated in 10 random fields of view for triplicate.

**Figure 2 ijms-25-00689-f002:**
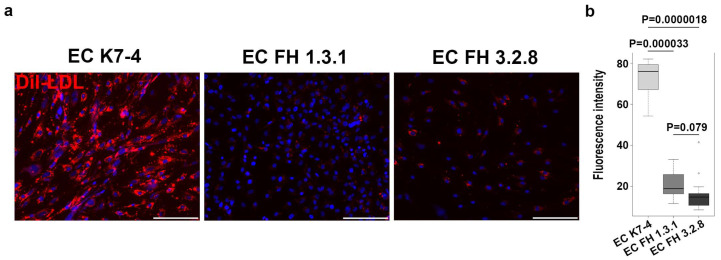
iPSC-derived endothelium from patients with FH shows reduced LDL uptake capacity. (**a**) Representative images showing differences in the fluorescently labeled LDL uptake capacity (red signal) between endothelial derivatives obtained from iPSCs with normal (EC K7-4) and pathological (EC FH 1.3.1 and EC FH 3.2.8) LDLR allelic variants. Nuclei are stained with DAPI. The scale bar is 100 microns. Magnification 200×. The experiment was performed in triplicate. (**b**) Quantification of LDL uptake capacity for control and patient-specific IPSC-derived endothelial cells in 10 fields of view for each sample.

**Figure 3 ijms-25-00689-f003:**
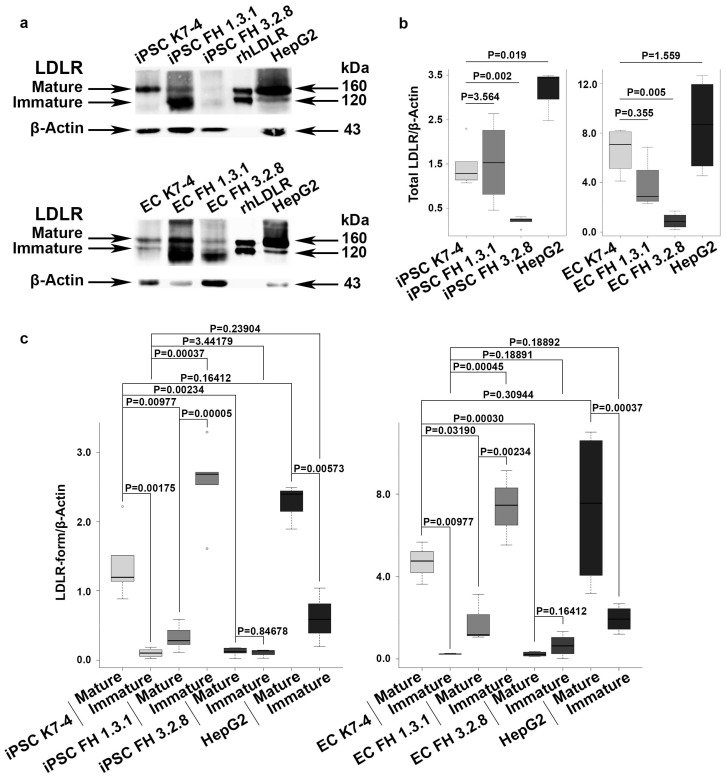
LDLR protein expression in iPSCs and their endothelial derivatives from patients with FH and a healthy donor. (**a**) Immunoblotting revealed a decreased level of mature LDLR in iPSCs and iPSC-ECs from patients with FH (FH 1.3.1 and FH 3.2.8) and an increased level of immature LDLR in iPSCs and iPSC-ECs from FH 1.3.1 compared to a healthy donor (K7-4). (**b**) Relative densitometric quantification of total LDL (mature and immature forms together) in iPSCs and iPSC-ES from patients with FH (FH 1.3.1 and FH 3.2.8) and a healthy donor (K7-4). The experiment was performed in triplicate. (**c**) Relative densitometric quantification of mature and immature LDL in iPSCs and iPSC-ES from patients with FH (FH 1.3.1 and FH 3.2.8) and a healthy donor (K7-4). The experiment was performed in triplicate.

**Figure 4 ijms-25-00689-f004:**
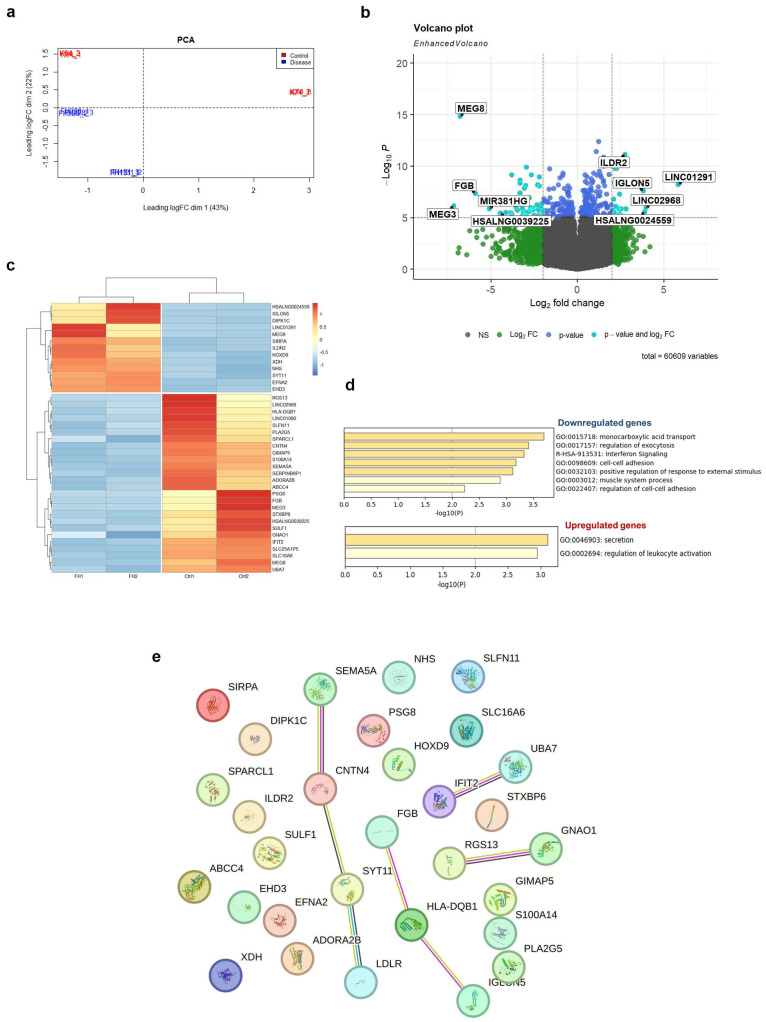
Transcriptional profiling of the control (CTRL) and *LDLR* mutant (FH) iPSC-ECs. (**a**) PCA plot for two control (CTRL) and two *LDLR* mutant (FH) iPSC-EC cultures. (**b**) Volcano plot of 39 differentially expressed genes (DEGs) between CTRL and FH. FDR, false discovery rate; FC, fold change. The top five DEGs are shown. (**c**) Heat map of 39 DEGs. The color of each dot represents the gene expression in the sample. The brighter the red, the higher the expression; the brighter the blue, the lower the expression. (**d**) Histograms showing gene ontology of biological processes and reactome gene sets with upregulated (logFC  ≥  1) and downregulated (logFC  ≤ −1) expression in FH compared to CTRL. (**e**) Bioinformatic analysis of DEGs using STRING to identify functional interactions between deregulated proteins. Each node represents a protein, and each edge represents an interaction.

**Table 1 ijms-25-00689-t001:** Primary and secondary antibodies.

Name	Supplier	Catalogue Number	RRID	Isotype	Dilution
**FACS**					
CD144 + PerCP-Cy5.5	BD	561566	AB_10715835	mouse IgG1	1/20
mouse IgG1 + PerCP-Cy5.5	BD	552834	AB_394484	mouse IgG1	1/50
**Immunofluorescence**					
CD 31	Cell Marque	131M-96	AB_1516765	mouse IgG1	1/50
Von Willebrand factor	Dako	A0082	AB_2315602	rabbit IgG	1/200
mouse IgG1 + Alexa 568	Thermo Fisher Scientific	A21124	AB_2535766	goat IgG	1/400
rabbit IgG + Alexa 488	Thermo Fisher Scientific	A110088	AB_143165	goat IgG	1/400
goat IgG + Alexa 488	Thermo Fisher Scientific	A11055	AB_2534102	donkey IgG	1/400
**Immunoblotting**					
LDLR	R&D System	AF2148	AB_2135126	goat IgG	1/1000
ACTB	Abcam	ab8227	AB_2305186	rabbit IgG	1/5000
goat IgG + peroxidase	Jackson ImmunoResearch	705-035-003	AB_2340390	donkey IgG	1/5000
rabbit IgG + peroxidase	Jackson ImmunoResearch	711-035-152	AB_10015282	donkey IgG	1/5000

## Data Availability

Raw RNA-seq data that supporting the results of this study are available from the corresponding author upon reasonable request.

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
