# Peer review of "iPSC-Derived Endothelial Cells Reveal LDLR Dysfunction and Dysregulated Gene Expression Profiles in Familial Hypercholesterolemia"

_ijms, 2024, doi:10.3390/ijms25020689_

Round 1
Reviewer 1 Report
Comments and Suggestions for Authors
The group Zakharova et al. study the defects in low-density lipoprotein receptor which are associated with familial hypercholesterolemia (FH) and cause different diseases including atherosclerosis and cardiovascular disease. In this study, the authors analyze the differences between endothelial cells from individuals with defective LDLR versus normal. The study found different genes which up or down regulated. The study is interesting, and the manuscript is gut but need few modifications which are mentioned down.
1- In the introduction, your statement “…coronary heart disease, and Alzheimer's disease.” Need references.
2- Also, the statement “…hepatocytes have impaired…. cholesterol levels in the bloodstream.” Need references.
3- Ref. 5 is not suitable, furthermore, one single reference is not enough you should cite suitable references in this position.
4- The sentence, “..It is believed that cellular metabolism and metabolites are crucial in regulating three-dimensional chromatin architecture and gene expression” should be rephrased to suit with the reference 6, or you have to cite other references because the reference 6 is about the lipid–gene interaction.
5- In the introduction, the authors should add more about the LDLR mutation and the complication caused
6- In the introduction, the authors must give more information about the cells used in study
7- You write in the introduction, that “…Therefore, we hypothesize that endothelial cells with dysfunctional LDLR and associated metabolic abnormalities in cellular metabolism…… extracellular cholesterol levels.”. Your hypotheses should be based on previous works. Here you should cite more works and references before you come to the hypothesis.
8- This part “we obtained two lines of induced pluripotent stem cells (iPSCs) from patients who were compound heterozygotes for pathogenic and likely pathogenic allelic variants of the LDLR gene…… and patients with FH” should be in the material and methods part not in the introduction
9- The part “We showed that typical…. cardiovascular disease in patients with FH” should be in the results and discussion part.
10- The introduction is poor, and not organized, it needs a radical rewriting.
11- In material and methods, for the mutation in the used cells, you should mention the rsnumber
12- In material and methods line 337 and 341, “…. Accutase enzyme (Thermo Fisher Scientific)” and “…Express (Thermo Fisher Scientific)”, you should mention the city and country as previously done. This must be uniform in the whole manuscript
13- Line 351 and 354 typos 105 and 104 must be corrected. Check the whole text for the same typos.
14- In the Immunostaining and other part. mention the camera used for imaging
15- In line 418 “Raw RNA-seq data that support the findings of this study are available from the corresponding author, upon reasonable request” should be deleted, this is a part of Data Availability Statement
16- Line 427-437 “RNA sequencing…. STRING database (https://string-db.org).” should be in RNA-seq part.
17- In Figure 1 and 2 add the magnification for the pictures
18- In the figure legends. add the number of replicates
19- The discussion part from line 202-225 should be normally in the introduction part.
20- In the discussion, the authors should cite the results they found and compare it with previous works.
21- Why you choose only two samples with limited mutations, the work could be more interesting if you add more mutation and more replicates from different patients with same mutation?
Comments on the Quality of English Language
Minor editing of English language required
Author Response
We thank the reviewer for the valuable comments. We have made changes that we believe have strengthened the manuscript, including removing accents and improving clarity. We have revised the manuscript significantly to meet the high standards for publication in IJMS.
Quality of English Language
(x) Minor editing of English language required
We did all our best to improve the English throughout the manuscript.
Does the introduction provide sufficient background and include all relevant references?
Must be improved
The introduction has been revised.
Are all the cited references relevant to the research?
Must be improved
All cited references have been verified and corrected.
1- In the introduction, your statement “…coronary heart disease, and Alzheimer's disease.” Need references.
We have added a reference for genome-wide association studies.
2- Also, the statement “…hepatocytes have impaired…. cholesterol levels in the bloodstream.” Need references.
We have moved up the references [2-4] to support the statement.
3- Ref. 5 is not suitable, furthermore, one single reference is not enough you should cite suitable references in this position.
The reference and the sentence to which it refers were removed when we rewrote the Introduction.
4- The sentence, “..It is believed that cellular metabolism and metabolites are crucial in regulating three-dimensional chromatin architecture and gene expression” should be rephrased to suit with the reference 6, or you have to cite other references because the reference 6 is about the lipid–gene interaction.
The reference is not just about direct lipid-gene interaction, but more about the regulatory role of lipids in RNA expression. It considers examples where chromatin structure and transcription are altered by changes in the levels of metabolites associated with lipid metabolism. It also includes how cholesterol is involved in transcriptional regulation. However, the reference with related text has been removed during editing of the Introduction.
5- In the introduction, the authors should add more about the LDLR mutation and the complication caused
We have added information about LDLR mutation and the complications it causes in the Introduction.
6- In the introduction, the authors must give more information about the cells used in the study
We have included text in the Introduction from the Discussion section describing disease modelling using iPSCs, as you recommended below.
7- You write in the introduction, that “…Therefore, we hypothesize that endothelial cells with dysfunctional LDLR and associated metabolic abnormalities in cellular metabolism…… extracellular cholesterol levels.”. Your hypotheses should be based on previous works. Here you should cite more works and references before you come to the hypothesis.
This text is no longer in the introduction.
8- This part “we obtained two lines of induced pluripotent stem cells (iPSCs) from patients who were compound heterozygotes for pathogenic and likely pathogenic allelic variants of the LDLR gene…… and patients with FH” should be in the material and methods part not in the introduction
We left this sentence in the Introduction to make it clear to readers that (1) the paper does not include the derivation and characterization of patient-specific iPSCs; (2) the patient-specific iPSCs we used do not contain normal LDLR alleles.
9- The part “We showed that typical…. cardiovascular disease in patients with FH” should be in the results and discussion part.
We have left a short summary of the results in the Introduction, as we used to do, especially since other reviewers do not mind.
10- The introduction is poor, and not organized, it needs a radical rewriting.
We have done our best to improve the introduction.
11- In material and methods, for the mutation in the used cells, you should mention the rsnumber
RS numbers (dbSNP IDs) have been added in section 4.2 'iPSC lines and their cultivation' of 'Materials and methods'.
12- In material and methods line 337 and 341, “…. Accutase enzyme (Thermo Fisher Scientific)” and “…Express (Thermo Fisher Scientific)”, you should mention the city and country as previously done. This must be uniform in the whole manuscript
We follow a pattern where the first reference to a company includes the city and country, and subsequent references include only the company name.
13- Line 351 and 354 typos 105 and 104 must be corrected. Check the whole text for the same typos.
We did all our best to improve the English throughout the manuscript.
14- In the Immunostaining and other part mention the camera used for imaging
We have added information about the camera in the Immunostaining and Endothelial Cell Function sections.
15- In line 418 “Raw RNA-seq data that support the findings of this study are available from the corresponding author, upon reasonable request” should be deleted, this is a part of Data Availability Statement
The statement has been removed from the RNA-seq methods section.
16- Line 427-437 “RNA sequencing…. STRING database (https://string-db.org).” should be in RNA-seq section.
RNA_seq data processing is in most cases based on statistical criteria and requires additional statistical tests from the R software package. Therefore, we decided to leave RNA_seq data processing in the 'Statistics' section.
17- In Figure 1 and 2 add the magnification for the pictures
Magnification has been added to Figures 1 and 2.
18- In the figure legends. add the number of replicates
The number of replicates has been added to Figures 1–3.
19- The discussion part from line 202-225 should be normally in the introduction part.
As recommended, this text has been moved to the Introduction.
20- In the discussion, the authors should cite the results they found and compare it with previous works.
Unfortunately, our work is the first and only study of how LDLR mutations affect gene expression in endothelial cells. There are no previous results to compare with our findings. Therefore, we discussed how our iPSC-EC could facilitate (contribute to) future FH research and how our findings on patient-specific iPSC-EC could contribute to the understanding of FH pathogenesis.
21- Why you choose only two samples with limited mutations, the work could be more interesting if you add more mutation and more replicates from different patients with same mutation?
We include in our study iPSC from genotyped compound heterozygote patients with the most severe clinical FН manifestations to see the most potent effects of LDLR mutations in endothelial cells. Unfortunately, our choice of compound heterozygote patients was severely limited. We hope that future work will allow us to extend this study.
Reviewer 2 Report
Comments and Suggestions for Authors
Materials and methods part is well-written.
It would be better to describe briefly FH 1.3.1S, FH 3.2.8T and K7-308 4Lf and K6-4f cell lines in Introduction part, as they were obtained in the previous study.
There is minor flaw in the Results:
In Figure 1 and Figure 2 should be useful add alphabet characters to the box plots.
- The Discussion should be rewritten.
Discussion is typically initiated by presenting the most significant findings of the study and then proceeding to their analysis. The first three paragraphs in your discussion are more appropriate for Introduction part.
- "Thus, the quantitative reduction of mature LDLR in iPSC-ECs abolished endothelial function, which was manifested both in the decreased ability to uptake LDL and at the transcriptomic level." This statement is not clear, as transcriptomic changes lead to changes in quantity of protein but not vice versa.
- Did you evaluate the expression level of PCSK9 gene in your cells? It might be interesting to add this information, as PCSK9 is a main regulator of LDLR.
- This is necessary to add reference to the description of the patient FH 3.2.8.
- "We found no difference in LDLR gene expression between FH and CTRL iPSC-ECs, suggesting that LDLR function is mainly abolished at the post-translational level, as also shown by immunoblotting results."
It is better to say about post-transcriptional changes than post-translational because of mutations in the LDLR gene. Otherwise, you should show that the are changes in glycosylation, phosphorilation etc.
Comments on the Quality of English LanguageThere is no word "metabolization". You should use "metabolism".
Line 45, 95, 121 LDL instead of low-density lipoprotein should be used, as this abbreviation was used above.
"Our study focused on identifying both the LDLR protein expression 59 and transcriptomic patterns in endothelial cells with pathogenic allelic variants in the 60 LDLR gene and discussed how this may potentially contribute to the FH progression." This sentence should be rephrased or divided into two sentences.
"The results that showed us the ability of iPSC-ECs to uptake low-density lipoproteins indicate that the obtained cells have LDL receptors." Please rephrase.
Line 238, the article is missed. It should be "In the patient".
Author Response
We thank the reviewer for the valuable comments. We have made changes that we believe have strengthened the manuscript, including removing accents and improving clarity. We have revised the manuscript significantly to meet the high standards for publication in IJMS.
Quality of English Language
(x) Minor editing of English language required
We did all our best to improve the English throughout the manuscript.
Does the introduction provide sufficient background and include all relevant references?
Must be improved
The introduction has been revised.
Are the results clearly presented?
Can be improved
The results has been revised.
It would be better to describe briefly FH 1.3.1S, FH 3.2.8T and K7-308 4Lf and K6-4f cell lines in Introduction part, as they were obtained in the previous study.
We have described the cell lines used in the work in detail in the 'Materials and methods', leaving the mention of the lines in the 'Introduction', although other reviewers have recommended that even this mention be moved to the 'Materials and methods'.
There is minor flaw in the Results:
In Figure 1 and Figure 2 should be useful add alphabet characters to the box plots.
Done as recommended.
- The Discussion should be rewritten.
Discussion is typically initiated by presenting the most significant findings of the study and then proceeding to their analysis. The first three paragraphs in your discussion are more appropriate for Introduction part.
As you suggested, we have moved the text describing disease modelling using iPSCs from the Discussion to the Introduction section. Unfortunately, our work is the first and only study of how LDLR mutations affect gene expression in endothelial cells. There are no previous results to compare with our findings. Therefore, we discussed how our iPSC-EC could facilitate (contribute to) future FH research and how our findings on patient-specific iPSC-EC could contribute to the understanding of FH pathogenesis. We did our best to improve the discussion.
- "Thus, the quantitative reduction of mature LDLR in iPSC-ECs abolished endothelial function, which was manifested both in the decreased ability to uptake LDL and at the transcriptomic level." This statement is not clear, as transcriptomic changes lead to changes in quantity of protein but not vice versa.
Let us disagree with you. Consider a situation where a pathological mutation affects a common transcription factor (TF). The gene for the TF has not changed its expression. However, due to the mutation, which leads to misfolding of the protein and its subsequent retention in the EPR, the level of this TF is reduced, so that the TF is unable to regulate (activate or repress) the expression levels of a large number of its target genes. This is a clear and obvious example of how changes in protein levels can lead to transcriptomic changes. LDLR is not a TF and its effect on the transcriptome is more complex and mediated by metabolism and metabolites. Metabolites such as S-adenosylmethionine, acetyl-CoA, used as substrates for DNA methylation and histone acetylation, regulate transcription (transcriptome) through the open or closed state of chromatin. A number of metabolites such as NAD+ and organic ions are involved in the regulation of chromatin modifying enzymes. In our work, we found that iPSC-ECs that lack normal LDLR alleles and have reduced levels of mature protein also have downregulated levels of monocarboxylic acid transporters, which directly and indirectly affect the levels of acetyl-CoA and organic ions involved in chromatin regulation. It should also be noted that metabolic pathways in cells and organisms are interconnected, so it cannot be excluded that changes in the monocarboxylic acid cycle may not affect other pathways involved in transcriptional regulation.
- Did you evaluate the expression level of PCSK9 gene in your cells? It might be interesting to add this information, as PCSK9 is a main regulator of LDLR.
We have added information about PCSK9 expression based on RNA-seq results. There is no significant difference in the level of PCSK9 expression in iPSCs and in ECs between patients and healthy controls. In addition, the results of preliminary genetic analysis of our patients did not reveal any pathogenic allelic variants in the PCSK9 gene, which also indirectly indicates that its function is not disturbed and therefore there is no additional contribution to LDLR protein function.
- This is necessary to add reference to the description of the patient FH 3.2.8.
Done as recommended.
- "We found no difference in LDLR gene expression between FH and CTRL iPSC-ECs, suggesting that LDLR function is mainly abolished at the post-translational level, as also shown by immunoblotting results."
It is better to say about post-transcriptional changes than post-translational because of mutations in the LDLR gene. Otherwise, you should show that the are changes in glycosylation, phosphorilation etc.
Done as recommended.
Comments on the Quality of English Language
There is no word "metabolization". You should use "metabolism".
The word "metabolization" exists and can be found in online dictionaries (https://en.wiktionary.org/wiki/metabolization or https://www.collinsdictionary.com/dictionary/english/metabolization ), but at your request we have not used it in the text.
Line 45, 95, 121 LDL instead of low-density lipoprotein should be used, as this abbreviation was used above.
Done as recommended.
"Our study focused on identifying both the LDLR protein expression 59 and transcriptomic patterns in endothelial cells with pathogenic allelic variants in the 60 LDLR gene and discussed how this may potentially contribute to the FH progression." This sentence should be rephrased or divided into two sentences.
We have rephrased this sentence.
"The results that showed us the ability of iPSC-ECs to uptake low-density lipoproteins indicate that the obtained cells have LDL receptors." Please rephrase.
We have also rephrased this sentence.
Line 238, the article is missed. It should be "In the patient".
We have fixed it.
Reviewer 3 Report
Comments and Suggestions for Authors
The paper by Zakharova et al investigates the functional defect in endothelial cells due to mutations in Low-density lipoprotein receptors (LDLR). Mutations and functional defects in LDLR receptors are shown to cause familial hypercholesterolemia. Using IPSc-derived endothelial cells from patients with LDLR mutations and healthy control, this study explores how the mutations in LDLR affect the function of endothelial cells and contribute to familial hypercholesteremia. Overall, the authors try to answer a very important question about how different cell types could contribute to the disease pathogenesis of familial hypercholesterolemia with differentiated endothelial cells as a model. However, I do have some comments.
Major comments
1. In Figure 1, the authors show cadherin-positive cells, it would be important to have co-staining images of CD38-positive endothelial cells with Von Willebrand factor and have better resolution images.
2. Authors should also mention experimental N numbers for FACS sorting and capillary-like structure formation Figure 1 C. Each analysis should mention the number of experiments or images used for making the graph.
3. In figure 2 legend should include n number of experiments and images used for quantification.
4. Figure 3 shows that differentiated EC cells from patients have reduced levels of mature LDLR which is also seen with reduced uptake of labeled LDL shown in Figure 2. Would it be possible to have lipid staining of EC cells to see if they have intracellular lipid accumulation or changes in lipid metabolism as compared to control cells?
5. RNA seq data shows changes that should be discussed more in terms of disease pathophysiology.
6. The study would greatly benefit from either metabolomic or lipidomics analysis if the authors had samples for these. Taking into consideration that these patients have functional defects in LDLR, metabolic changes in pathways would give a better understanding of affected pathways.
Minor comments
1. The figure legends should include experimental n numbers.
2. Figure 4 has volcano plot with red and green colors together might be challenging for some people.
3. A thorough reading for grammar and language editing would benefit the manuscript.
Comments on the Quality of English LanguageThe manuscript will benefit from thorough English editing for grammar and flow of sentences.
Author Response
We thank the reviewer for the valuable comments. We have made changes that we believe have strengthened the manuscript, including removing accents and improving clarity. We have revised the manuscript significantly to meet the high standards for publication in IJMS.
Quality of English Language
(x) Moderate editing of English language required
We did all our best to improve the English throughout the manuscript.
Does the introduction provide sufficient background and include all relevant references?
Can be improved
The introduction has been revised.
Is the research design appropriate?
Can be improved
We have added new information to the experimental results. Several proposed experiments will be conducted as part of the ongoing research.
Are the results clearly presented?
Can be improved
The presentation of the results has been improved.
Are the conclusions supported by the results?
Can be improved
The conclusions have been revised.
Major comments
- In Figure 1, the authors show cadherin-positive cells, it would be important to have co-staining images of CD38-positive endothelial cells with Von Willebrand factor and have better resolution images.
We believe that the reviewer was referring to the co-staining of endothelial cells with CD31 (not CD38) and von Willebrand factor, as CD38 is not a typical endothelial marker.
We co-stained endothelial cells with CD31 and von Willebrand factor, and the resulting images are shown in Supplementary Figure 1 (please see the attachment).
In response to the comments about image resolution, we note that the original high quality images can be found at a link https://figshare.com/projects/iPSC-derived_endothelial_cells_reveals_LDLR-dysfunction_and_dysregulated_gene_expression_profiles_in_familial_hypercholes-terolemia/189747 .
Unfortunately, when the images are placed in a *doc file during manuscript preparation, the resolution became poorer. During paper production, the publisher usually uses the original high resolution images.
- Authors should also mention experimental N numbers for FACS sorting and capillary-like structure formation Figure 1 C. Each analysis should mention the number of experiments or images used for making the graph.
Done as recommended.
- In figure 2 legend should include n number of experiments and images used for quantification.
Information on experimental replicates and images used for quantification are included in the figure legends.
- Figure 3 shows that differentiated EC cells from patients have reduced levels of mature LDLR which is also seen with reduced uptake of labeled LDL shown in Figure 2. Would it be possible to have lipid staining of EC cells to see if they have intracellular lipid accumulation or changes in lipid metabolism as compared to control cells?
Oil red lipid staining showed no lipid inclusions in LDL-treated ECs (Please see the attachment or https://figshare.com/s/5c6f630ffaf6cefe4b33). As a positive control, we used hepatocyte-like cells differentiated from hPSCs of a patient with FH. The negative results may be due to inappropriate experimental conditions and other reasons.
For example, lipid droplet formation depends on the presence of serum in the growth medium, the concentration and the duration of lipids treatment (at least 24 hours) (https://doi.org/10.3390/cells10061403). Therefore, the experiment requires careful selection of conditions. Furthermore, the detection of lipid droplets requires the use of high spatial resolution images (Raman microscopy).
Actually, intracellular lipid accumulation in endothelial cells has been shown to occur in response to inflammation induced by LPS treatment (https://doi.org/10.3390/cells10061403) and TNF-stimulated inflammation (https://doi.org/10.1038/srep40889).
However, the mechanisms and pathways for the biogenesis of lipid droplets formation in the endothelium, especially during inflammation, remain incompletely identified (https://doi.org/10.1007/s00018-022-04362-7, doi: 10.1161/CIRCRESAHA.116.310498).
The pathways of lipid entry into ECs are not limited to the LDLR-mediated mechanism. In particular, as plasma LDL-C levels rise, ECs are exposed to oxidised LDL, which is internalised by ECs with the involvement of the LOX-1 receptor (doi: 10.3389/fcvm.2022.925923).
Overall, we are interested in this question and plan to conduct a separate study on the involvement of LDL metabolism of EC in patients with FH using Raman microscopy.
- RNA seq data shows changes that should be discussed more in terms of disease pathophysiology.
We have discussed RNA seq data in relation to disease pathophysiology.
- The study would greatly benefit from either metabolomic or lipidomics analysis if the authors had samples for these. Taking into consideration that these patients have functional defects in LDLR, metabolic changes in pathways would give a better understanding of affected pathways.
We have planned separate work to investigate the metabolomic and lipidomic profiles of iPSC-derived endothelial cells from FH patients. This will be a separate experimental study. In order to see differences in the metabolomic and lipidomic profiles of endothelial cells, it is necessary to expose them to oxidised LDL. In addition, these experiments are better performed on isogenic cell lines to eliminate the contribution of genetic background. Such work is currently underway in the laboratory and will be presented as a separate publication.
Minor comments
- The figure legends should include experimental n numbers.
Information on experimental replicates has been included in the figure legends.
- Figure 4 has volcano plot with red and green colors together might be challenging for some people.
In Figure 4, we replaced the red color with a bright turquoise. The modified image was shown to people with protanopia and they were able to easily distinguish the colors in the plot.
- A thorough reading for grammar and language editing would benefit the manuscript.
Comments on the Quality of English Language
The manuscript will benefit from thorough English editing for grammar and flow of sentences.
We did all our best to improve the English throughout the manuscript.

Round 2
Reviewer 1 Report
Comments and Suggestions for Authors
1-To the response 1, and also the other responses the author must mention exactly where he makes the modification (line number) highlighting is not enough as there are other changes made based on suggestions of other reviewers, otherwise, it takes longer and it is difficult to found where the modifications are made.
2-To Response 9, the results should not be included in the introduction, summary of the results should be in the abstract, and the rest in the result part. I never see paper mention the results in the introduction, and each reviewer has own decision and he is responsible for that.
3- To response 12, I write and review many manuscripts and always the kits/reagents were mentioned with company, city and country. The reason is that some products are available in some countries but not in others.
4- To response 13, the typos are still there, (Line 386 and 389) 105 cells are not 100000, and 104 cells are not 10000, you should take the reviewer comments seriously.
5- To response 17, the magnification you can mention in the figure legend, but it is more favorable if you add it in the figure.
Author Response
Dear Editor and Reviewer, we have responded to the second round of comments and hope that the manuscript has improved further.
1-To the response 1, and also the other responses the author must mention exactly where he makes the modification (line number) highlighting is not enough as there are other changes made based on suggestions of other reviewers, otherwise, it takes longer and it is difficult to found where the modifications are made.
Response 1 (round 1): line 45 (ref. [6])
Response 2 (round 1): lines 48-49 (ref. 3,10)
Response 3 (round 1): The reference and the sentence to which it refers were removed when we rewrote the Introduction (lines 36-97).
Response 4 (round 1): <...> the reference with related text has been removed during editing of the Introduction.
Response 5 (round 1): lines 39-43, 49-51, 53-55.
Response 6 (round 1): lines 61-91.
Response 7 (round 1): This text is no longer in the introduction.
Response 8 (round 1): lines 89-93.
Response 9 (round 1): lines 483-493 (response 2, round 2 – see below).
Response 10 (round 1): lines 36-97.
Response 11 (round 1): lines 336-338.
Response 12 (round 1): lines 376-377, lines 372-373, line 386 (response 3, round 2 – see below).
Response 13 (round 1): lines 389, 393, 411, 428 (response 4, round 2 – see below).
Response 14 (round 1): lines 402-403, 415.
Response 15 (round 1): lines 515-516.
Response 16 (round 1): lines 480-481.
Response 17 (round 1): lines 143, 145, 154, 498, response 5, round 2 – see below.
Response 18 (round 1): lines 140-141, 147, 154, 156, 198-199, 200-201.
Response 19 (round 1): lines 61-91.
Response 20 (round 1): lines 245-248, 253-255, 257-259, 261-269, 275-279, 280-293, 302-305, 306-326.
Response 21 (round 1): lines 86-91.
2-To Response 9, the results should not be included in the introduction, summary of the results should be in the abstract, and the rest in the result part. I never see paper mention the results in the introduction, and each reviewer has own decision and he is responsible for that.
We have followed your recommendation and only mentioned our conclusion in the final.
We also provide references to articles in the International Journal of Molecular Sciences and other academic journals that describe their results in the last paragraph of the Introduction. The references are given below.
https://doi.org/10.3390/ijms241411860
doi: 10.1016/j.stemcr.2019.01.017
doi:10.1242/dmm.042911
doi: 10.1002/hep4.1110
https://doi.org/10.1016/j.stem.2020.11.003
https://doi.org/10.3390/ijms24054471
https://doi.org/10.3389/fbioe.2022.772981
3- To response 12, I write and review many manuscripts and always the kits/reagents were mentioned with company, city and country. The reason is that some products are available in some countries but not in others.
We had not faced such a presentation of reagents before and misunderstood its format, especially as there are no clear guidelines in the JMS Instructions for Authors. In the second round of peer review, it became clear to us what you were talking about, and we corrected the references to reagent manufacturers according to the format you would expect.
Lines 376-377: TrypLE Express (Thermo Fisher Scientific, Waltham, MA, USA),
Lines 372-373: Accutase enzyme (Thermo Fisher Scientific, Waltham, MA, USA),
Line 386: Accutase enzyme (Thermo Fisher Scientific, Waltham, MA, USA).
4- To response 13, the typos are still there, (Line 386 and 389) 105 cells are not 100000, and 104 cells are not 10000, you should take the reviewer comments seriously.
We made corrections to the cell numbers in the text. In particular, we changed '105' to '10^5' on line 389 and '104' to '10^4' on line 393. To ensure consistency throughout the text, we also replaced '105' with '10^5' on lines 411 and 428.
5- To response 17, the magnification you can mention in the figure legend, but it is more favorable if you add it in the figure.
In the realities of modern digital technology and imaging, it is more relevant to use a scale bar rather than a magnification. The magnification of an image is only relevant if you have a 1:1 printed image on paper. If you are viewing an electronic version of the image, the actual magnification will depend on the device and zoom options. For example, an image viewed on a large computer monitor will have a higher magnification than the same image viewed on a smartphone screen. However, the caption on the picture says that both the larger and smaller versions have the same magnification. Do you realize that this is incorrect? That is why a scale bar (not a magnification) is placed on the images to allow you to accurately judge the size of microscopic objects.
In line with this, we have included the magnification in the figure legends to show our respect for tradition, but have not added it directly to the image, as it has no meaning with the digital version of the image. Moreover, the IJMS only has an electronic but not a print version.
The JMS Instructions for Authors (Preparing Figures, Schemes and Tables https://www.mdpi.com/journal/ijms/instructions#references) do not require to include magnification in figures. In response to reviewer 1's request during the first peer review round, we mentioned magnification in figure legends. We believe that it is not correct to add magnification directly to the images and that it is more appropriate to use the scale bars.